# Mitochondrial Dysfunction in Spinocerebellar Ataxia Type 3 Is Linked to VDAC1 Deubiquitination

**DOI:** 10.3390/ijms23115933

**Published:** 2022-05-25

**Authors:** Tina Harmuth, Jonasz J. Weber, Anna J. Zimmer, Anna S. Sowa, Jana Schmidt, Julia C. Fitzgerald, Ludger Schöls, Olaf Riess, Jeannette Hübener-Schmid

**Affiliations:** 1Institute of Medical Genetics and Applied Genomics, University of Tübingen, 72076 Tübingen, Germany; tina.harmuth@med.uni-tuebingen.de (T.H.); jonasz.weber@med.uni-tuebingen.de (J.J.W.); annazimmer84@gmail.com (A.J.Z.); anna.s.sowa@gmail.com (A.S.S.); j.schmidt@med.uni-tuebingen.de (J.S.); olaf.riess@med.uni-tuebingen.de (O.R.); 2Centre for Rare Diseases, University of Tübingen, 72076 Tübingen, Germany; 3Graduate School of Cellular Neuroscience, University of Tübingen, 72076 Tübingen, Germany; 4Department of Human Genetics, Ruhr University Bochum, 44801 Bochum, Germany; 5Centre of Neurology, Department of Neurodegenerative Diseases, Hertie Institute for Clinical Brain Research, University of Tübingen, 72076 Tübingen, Germany; julia.fitzgerald@uni-tuebingen.de; 6Department of Neurodegenerative Diseases, Hertie Institute for Clinical Brain Research, University of Tübingen, 72076 Tübingen, Germany; ludger.schoels@uni-tuebingen.de; 7German Centre for Neurodegenerative Diseases (DZNE), 72076 Tübingen, Germany

**Keywords:** ataxin-3, Machado–Joseph disease, mitochondria dysfunction, spinocerebellar ataxia type 3, VDAC1 ubiquitination

## Abstract

Dysfunctional mitochondria are linked to several neurodegenerative diseases. Metabolic defects, a symptom which can result from dysfunctional mitochondria, are also present in spinocerebellar ataxia type 3 (SCA3), also known as Machado–Joseph disease, the most frequent, dominantly inherited neurodegenerative ataxia worldwide. Mitochondrial dysfunction has been reported for several neurodegenerative disorders and ataxin-3 is known to deubiquitinylate parkin, a key protein required for canonical mitophagy. In this study, we analyzed mitochondrial function and mitophagy in a patient-derived SCA3 cell model. Human fibroblast lines isolated from SCA3 patients were immortalized and characterized. SCA3 patient fibroblasts revealed circular, ring-shaped mitochondria and featured reduced OXPHOS complexes, ATP production and cell viability. We show that wildtype ataxin-3 deubiquitinates VDAC1 (voltage-dependent anion channel 1), a member of the mitochondrial permeability transition pore and a parkin substrate. In SCA3 patients, VDAC1 deubiquitination and parkin recruitment to the depolarized mitochondria is inhibited. Increased p62-linked mitophagy, autophagosome formation and autophagy is observed under disease conditions, which is in line with mitochondrial fission. SCA3 fibroblast lines demonstrated a mitochondrial phenotype and dysregulation of parkin-VDAC1-mediated mitophagy, thereby promoting mitochondrial quality control via alternative pathways.

## 1. Introduction

Neuronal function is highly dependent on mitochondria as neurons have a high energy demand. Therefore, it is not surprising that most neurodegenerative diseases are linked to impaired mitochondrial function. Symptoms resulting from dysfunctional mitochondria, such as metabolic defects and body weight loss, can be observed as early symptoms in patients with neurodegenerative diseases [1,2]. Spinocerebellar ataxia type 3 (SCA3), also known as Machado–Joseph disease (MJD), is the most frequent, dominantly inherited ataxia worldwide and was recently linked to mitochondrial dysfunction and impaired mitophagy [3,4,5,6,7]. SCA3 is caused by a CAG repeat expansion in the *MJD1* gene that leads to an expanded polyglutamine (polyQ) tract in the encoded ataxin-3 protein, which makes SCA3 belong to the polyQ-disease family. Ataxin-3 functions as a deubiquitinating enzyme in protein quality control and preferentially cleaves mixed linked K48-K63 polyubiquitin chains from substrates [8]. One known substrate of ataxin-3 is parkin (encoded by *PARK2*) [9,10,11], an E3 ubiquitin-protein ligase and key protein in mitochondrial quality control that is frequently mutated in patients with inherited juvenile Parkinson’s disease (PD). It ubiquitinates mitochondrial outer membrane proteins such as VDAC1 (voltage-dependent anion channel 1) and mitofusins, and thereby recruits autophagic proteins such as p62 to induce degradation of dysfunctional mitochondria by mitophagy [12]. This cell-protective process, known as canonical parkin-mediated mitophagy, can be impaired by increased autophagic clearance of parkin upon ubiquitination. The fact that direct ubiquitination of parkin is enhanced in the presence of an elongated polyQ stretch may explain neuronal cell loss and PD-like neurological symptoms of some SCA3 patients [9,10,11,13,14,15]. Furthermore, the mitochondrial phenotype in SCA3 is various and, thus, offers other promising points of action explaining neuronal cell loss. Mitochondrial damage indicated by increased mitochondrial DNA (mtDNA) deletions in symptomatic and preclinical SCA3 patients as well as mtDNA deletions and decreased mitochondrial copy numbers in SCA3 mouse models were reported [16,17,18]. Ataxin-3 itself was shown to localize to mitochondria [19,20], and a calpain-mediated cleavage fragment of ataxin-3 is potentially responsible for mitochondrial fragmentation in SCA3 [7,21]. Additionally, mass spectrometry analyses revealed new mitochondrial proteins as potential interaction partners of ataxin-3 under normal and also disease conditions [20]. Proteome analyses of an SCA3 knockin mouse model demonstrated energy metabolism and dysfunctional expression of mitochondrial proteins as an early event in the pathogenesis [22]. Therapeutically, vitamin B6 supplementation in an SCA3 Drosophila model resulted in mitochondrial protection of polyQ-induced cellular toxicity [23].

In our current study, we analyzed mitochondrial dysfunction and levels of mitophagy in an immortalized SCA3 patient-derived fibroblast cell line (iHF) and confirmed some of our findings in a mouse embryonic fibroblast cell line isolated from SCA3 transgenic mice expressing human ataxin-3 with 148Q (MEF 148Q) [24]. SCA3 patient-derived fibroblasts exhibited abnormalities in mitochondrial morphology that were linked to reduced protein expression in mitochondrial fusion proteins. Additionally, differential ubiquitination of mitochondrial permeability transition pore protein VDAC1 was found in SCA3 patient-derived fibroblasts, showing even stronger effects in the MEF 148Q cell model. Using an in vitro deubiquitination assay we demonstrated that wildtype ataxin-3 directly deubiquitinates VDAC1. Furthermore, the mitochondrial phenotype in SCA3 patient-derived fibroblasts leads to increased mitophagy, suggesting that loss of mitochondrial quality induces mitochondrial renewal. In summary, our study presents for the first time that wildtype ataxin-3 directly deubiquitinates VDAC1 and that mitochondrial dysfunction potentially directly influenced by polyQ-expanded ataxin-3 promotes increased mitochondrial renewal/mitophagy.

## 2. Results

### 2.1. Mitochondrial Morphology Is Altered in SCA3 Cell Models

In different cell models of SCA3, fragmented and less interconnected mitochondria were detected [7,20,21]. To confirm these features in two further SCA3 cell models, mitochondrial morphology was investigated in immortalized human SCA3 fibroblasts (iHF SCA3) and mouse embryonic fibroblasts (MEF) overexpressing ataxin-3 with 148Q under control of the huntingtin promotor (MEF 148Q) [24] compared to their respective controls (iHF hc, MEF wt/wt). Both SCA3 cell models presented ring-shaped/doughnut-like mitochondria (Figure 1A,B). Additionally, in MEF 148Q mitochondria were condensed and clustered around the nucleus in 80–90% of all stained cells with an intact nucleus (Figure 1B). Because of the accumulation of mitochondria around the nucleus (MEF 148Q) and their spatial condensation (iHF SCA3, MEF 148Q), it was not possible to evaluate the mitochondrial morphology with Image J. The software was not able to discriminate single mitochondrion and counted several mitochondria in close proximity as one large mitochondrion (data not shown).

Changes in mitochondrial morphology can be regulated by a differential expression of mitochondrial fusion and fission proteins. Therefore, Western blot analyses were performed for mitochondrial fusion proteins, mitofusin 2 (MFN2) and for the mitochondrial fission protein 1 (FIS1). Additionally, optic atrophy 1 (L(ong)- and S(hort)-OPA1), a dynamin-related GTPase that controls mitochondrial dynamics, cristae integrity, energetics and mtDNA maintenance, was evaluated. The membrane-anchored L-OPA1 determines fusion and a long or soluble S-OPA1 alone builds cristae, whereas long and short forms together tune mitochondrial morphology [25]. In line with observed mitochondrial morphological changes, SCA3 patient-derived fibroblasts demonstrated reduced membrane-anchored L- and soluble S-OPA1 protein levels (Figure 1C–E; L-OPA1 *p* = 0.002; S-OPA1 *p* = 0.049) and significantly reduced MFN2 protein levels (*p* = 0.0026, Figure 1C,F). The fission protein FIS1 was unchanged (Figure 1C,G).

### 2.2. PolyQ-Expanded Ataxin-3 Affects Proper Mitochondrial Function

Electron microscopy of iHF and MEF revealed an intact inner mitochondrial membrane (IMM) and cristae structure in cells expressing polyQ-expanded ataxin-3, compared to controls. No conclusion about mitochondria length was possible by electron microscopy, since we do not know where the section plan was exactly located in the mitochondrion (Appendix A). The IMM harbors components required for oxidative phosphorylation (OXPHOS) and ATP-production. To evaluate repercussions of the functional integrity of the IMM, mitochondrial membrane potential, ATP production and protein expression of OXPHOS proteins were analyzed in iHF SCA3.

Mitochondrial membrane potential measured by TMRE staining followed by fluorescence-activated cell sorting (FACS) showed no significant changes in mitochondrial membrane potential in iHF SCA3 (Figure 2A). Nonetheless, the appearance of a second cell population in the FACS analysis for the majority of iHF SCA3, but only for some control samples (iHF hc), points to the existence of two distinct subgroups of cells in both cell lines (Appendix A). Additionally, analyses of ATP production by a luciferase-based immunoassay demonstrated significantly reduced ATP production in iHF SCA3 (*p* = 0.01, Figure 2B) compared to fibroblasts from a healthy control (iHF hc). PrestoBlue Cell Viability assays indicated a significantly reduced cell viability in iHF SCA3 compared to the fibroblasts from a healthy control (*p* < 0.001, Figure 2C).

Reduced ATP production in iHF SCA3 could be explained by alterations in protein expression of components of OXPHOS. Western blot analyses of OXPHOS protein subunits covering complexes I–V (Figure 2D) revealed significantly reduced protein levels for ubiquinol-cytochrome c reductase core protein II (UQCRC, CIII, *p* = 0.041, Figure 2F) and succinate dehydrogenase complex, subunit B (SDHB, CII, *p* = 0.034, Figure 2G). The analyzed subunits of complex I (NDUFB, Figure 2H) and complex V (ATP5a, Figure 2E) were unchanged in iHF SCA3 compared to iHF hc.

Oxidative phosphorylation and ATP production require a stable proton gradient across the inner mitochondrial membrane, built by complexes I-IV of the respiratory chain. Therefore, reduced ATP production and alterations in the OXPHOS protein expression with an implication on viability are first indications for dysfunctional mitochondria in our patient-based SCA3 cell model.

### 2.3. Parkin Is Not Recruited to Depolarized Mitochondria in SCA3 Patient-Derived Fibroblasts

The mitochondrial Ser/Thr protein kinase PINK1 (PTEN-induced putative kinase 1) and the E3 ubiquitin-protein ligase parkin (encoded by *PARK2*), both linked to familial PD, are known to accumulate in depolarized mitochondria. In healthy mitochondria, PINK1 is immediately cleaved between aa Ala103 and Phe104 (∆1) and degraded. In depolarized, damaged mitochondria, autophosphorylated PINK1 accumulates in the outer mitochondrial membrane (OMM) and recruits cytosolic parkin. Parkin ubiquitinates OMM-located proteins and labels them, and thereby the entire mitochondrion, for degradation by the proteasome or autophagy [26]. Interestingly, a direct interaction between ataxin-3 and parkin was shown and polyQ-expanded ataxin-3 was linked to an abnormal loss of parkin, which might be responsible for some rare parkinsonian features in SCA3 patients [9,10,11,13,14,15]. Therefore, we analyzed the PINK1 and parkin expression under normal growth conditions, but also after a 6 h treatment with 50 µM carbonyl cyanide m-chlorophenyl hydrazone (CCCP) to depolarize the mitochondria. Upon CCCP treatment, the previously detected ring-shaped mitochondrial morphology was enhanced in SCA3 patient-derived fibroblasts (Figure 3G). Western blot analyses of PINK1 under normal growth conditions revealed significantly reduced levels of cleaved ∆1-PINK1 in iHF SCA3 compared to healthy controls (Figure 3A,C, *p* = 0.005). After CCCP treatment, both ∆1- and full-length PINK1 accumulated in iHF SCA3 and iHF hc (Figure 3B,C, *p* = 0.8). Additionally, under normal growth conditions parkin showed very low, but comparable protein levels in all analyzed genotypes. As expected, in healthy controls (iHF hc), parkin expression was enhanced by CCCP treatment (*p* = 0.03), but this was not observed in the patient-derived iHF SCA3 line (Figure 3A,D). Confirmation of the observed effects on PINK1 and parkin protein levels in MEF 148Q and wildtype MEF failed due to generally very low expression levels of parkin in mouse embryonic fibroblasts (Appendix A). Additionally, PINK1 expression analyses revealed no significant differences between genotypes and treatments (Appendix A). Western blot analyses of the mitochondrial translocase of the outer membrane TOM20, which is ubiquitinated by parkin in depolarized mitochondria [27], revealed reduced protein levels in untreated SCA3 patient-derived fibroblasts (iHF SCA3, *p* = 0.07). TOM20 levels in CCCP-treated fibroblasts from both healthy controls and SCA3 patient-derived fibroblasts were also lower compared to untreated fibroblasts from healthy controls (*p* = 0.006 in CCCP iHF hc; *p* = 0.01 in CCCP iHF SCA3, Figure 3A,E). Similar expression levels were found for the mitochondrial marker citrate synthase (CS) independently from genotype and treatment. CS expression levels were only slightly reduced in CCCP-treated fibroblasts of healthy controls compared to untreated healthy controls (*p* = 0.08, Figure 3A,F). In MEF, Western blot analyses demonstrated higher CS expression in CCCP-treated wildtype MEF compared to untreated controls (*p* = 0.0017). On the other hand, CS expression levels in CCCP-treated MEF 148Q cells were unchanged compared to untreated MEF 148Q (Appendix A).

### 2.4. Ataxin-3 Deubiquitinates VDAC1 In Vitro

The permeability of OMM depends on membrane-crossing channel proteins. One of them is the voltage-dependent anion channel VDAC1, which allows small metabolites to freely diffuse across the outer mitochondrial membrane. Upon increased intracellular calcium concentrations and oxidative stress, VDAC1, together with other proteins, forms the mitochondrial permeability transition pore, which allows the unselected transition of solutes up to a size of 1.5 kDa and, therefore, leads to mitochondrial depolarization and impairment of ATP synthesis [28]. Additionally, it was shown that VDAC1 ubiquitination is the critical step in the regulation of mitophagy and autophagy [29]. In the next step, we analyzed total and ubiquitinated VDAC1 protein levels by Western blotting under normal growth conditions as well as under conditions of depolarized mitochondria after CCCP treatment (50 µM for 6 h). SCA3 iHF demonstrated an increase in full-length VDAC1 compared to the respective controls under normal growth conditions (Figure 4A,B, *p* = 0.089). Under CCCP treatment, the total full-length VDAC1 was reduced in both iHF SCA3 and the respective control (iHF hc, Figure 4A,B). One monoubiquitinated and two forms of polyubiquitinated VDAC1 (2,3) were detected in our Western blot analyses. Patient-derived iHF demonstrated a slight non-significant reduction in the ratio of monoubiquitinated or polyubiquitinated VDAC1 to the full-length VDAC1 protein level under normal growth conditions (Figure 4C–E). Upon CCCP treatment, a significant reduction in the ratio of mono- and polyubiquitinated VDAC1 was observed (Figure 4C–E, *p* = 0.0078 for mono-, and *p* = 0.0031 and *p* < 0.001 for polyubiquitinated VDAC1 forms 2 and 3, respectively).

Analyses in our second transgenic mouse-based cell model confirmed the effects in (ubiquitinated) VDAC1 protein levels. Under normal growth conditions an increase in total VDAC1 was found in MEF 148Q compared to the respective control (*p* = 0.0027). Under CCCP treatment, an increased total VDAC1 level was detected in both MEF wt/wt and MEF 148Q cells (Appendix A). The ratio of monoubiquitinated VDAC1 was reduced in MEF 148Q under normal growth conditions (*p* = 0.0065) and after CCCP treatment in both wildtype and MEF 148Q conditions (*p* < 0.01). The ratio of VDAC1 polyubiquitinated form 2 was significantly increased in MEF 148Q compared to the respective control under normal growth conditions (*p* = 0.0242), but comparable to CCCP-treated wildtype MEF (Appendix A). For polyubiquitinated form 3, only wildtype cells revealed a significant reduction after CCCP treatment (*p* = 0.0214; Appendix A).

It is known that VDAC1 is a target for parkin-mediated Lys27 polyubiquitination [30] and that the deubiquitinase ataxin-3 directly interacts with parkin [9,10,11]. As our data showed differences in expression of total and ubiquitinated forms of VDAC1 in our cell models, we raised the question whether this is caused by alterations in parkin-mediated ubiquitination or by ataxin-3-dependent deubiquitination of VDAC1. As shown earlier, patient-derived fibroblasts failed to recruit parkin to depolarized mitochondria, but showed comparable levels under normal growth conditions to their respective controls (Figure 3). To study the direct influence of ataxin-3 on VDAC1 ubiquitination, we performed an in vitro deubiquitination assay by incubating protein extracts from iHF SCA3 as well as iHF control samples (iHF hc) with exogenous His_6_-tagged ataxin-3. Western blot analyses revealed a time-dependent reduction in low- and high-molecular polyubiquitinated VDAC1 in both genotypes, with a stronger reduction in patient-derived fibroblasts (Figure 5A–D). For assay control, deubiquitination of the known ataxin-3 substrate p53 was analyzed (Figure 5E), as well as the lowering of K63-linked polyubiquitinated proteins (Figure 5F). Both demonstrated a time-dependent lowering of high-molecular (hmw) polyubiquitinated (pUb) forms of p53 as well as of K63-linked polyubiquitin chains, whereas medium- (mmw) and low-molecular (lmw) pUb forms of p53 were increased (Appendix A). Full-length protein levels of VDAC1 or p53 revealed no expression differences between both genotypes over time (Appendix A). As loading control GAPDH is shown, Ponceau S served as an additional loading control and also monitored protein integrity (Figure 5G). This data confirmed that ataxin-3 itself can deubiquitinate VDAC1 in vitro.

### 2.5. Mitochondrial Dysfunction Is Linked to Dysregulated Mitophagy in SCA3 Cells

VDAC1 is not only a component of the mitochondrial permeability transition pore, but also serves as a marker for mitochondrial-linked autophagy, or so-called mitophagy [30]. To investigate whether mitochondrial stress in patient-derived fibroblasts (iHF SCA3) influences mitophagy, we analyzed two important autophagic markers, namely, p62 and LC3. All analyses were performed under normal growth conditions and upon mitochondrial depolarization by CCCP treatment. Western blot analyses of p62 and LC3 revealed a similar ratio of LC3-II/LC3-I and similar p62 protein levels in iHF SCA3 compared to healthy controls under normal growth conditions (Figure 6A–C). After CCCP treatment, p62 protein levels were significantly elevated in iHF SCA3 compared to healthy controls (Figure 6A,B; *p* = 0.014). Additionally, CCCP treatment led to an increased ratio of LC3-II/LC3-I in both iHF hc and IHF SCA3, but was more effective in iHF SCA3 (for iHF hc *p* = 0.0011, for iHF SCA3 *p* < 0.001; Figure 6A,C).

To further investigate mitophagy, autophagosome formation was analyzed. For this, iHF were co-transfected with pEGPF-LC3 and pDSRed2-Mito vectors. For autophagy induction, cells were treated with rapamycin and, in parallel, the degradation of autophagosomes was inhibited by treatment with bafilomycin. Immunofluorescence microscopy revealed low numbers of autophagosomes in both iHF hc and iHF SCA3 under normal growth conditions (Figure 7A,C). Autophagy induction increased the number of autophagosomes significantly in both genotypes, with significantly higher autophagosome numbers under disease conditions (*p* = 0.0045). As double-immunofluorescence analyses revealed a close proximity between mitochondria and autophagosomes (Figure 7B), electron microscopy was performed to investigate whether autophagosomes contain mitochondria. Electron microscopy revealed round structures surrounded by a double membrane, potentially presenting autophagosomes in both genotypes and under normal growth conditions as well as under CCCP treatment. However, the cargo of these autophagosomes could not be clearly identified (Appendix A).

Western blot analyses of SCA3 patient-derived fibroblasts under rapamycin and bafilomycin treatments revealed similar PINK1 protein expressions in all analyzed conditions and only a slightly reduced parkin expression in iHF SCA3 cells under autophagy induction by rapamycin, compared to healthy controls (Figure 8A–C, *p* = 0.054). Additionally, the mitochondrial marker citrate synthase (CS) was significantly reduced in rapamycin-treated SCA3 patient-derived fibroblasts (*p* = 0.009) and increased after a combined rapamycin and bafilomycin treatment (*p* = 0.0263). In control fibroblasts, CS expression was not altered under rapamycin or rapamycin/ bafilomycin treatment (Figure 8A,D). The autophagic marker p62 demonstrated a significantly increased expression after CCCP treatment in both genotypes (Figure 8A,E). Additionally, the ratio of LC3-II/LC3-I revealed a significant induction of autophagy only in iHF SCA3 after CCCP treatment (Figure 8A,F, *p* = 0.0019).

In summary, the analyzed SCA3 patient-derived fibroblast model exhibited doughnut-like mitochondria accompanied by reduced expression of the fusion proteins MFN2 and the membrane-anchored L-OPA1. Moreover, the soluble S-OPA1, which builds cristae and, together with L-OPA1, tunes mitochondrial morphology, demonstrated reduced protein levels. Additionally, reduced cell viability and ATP production, as well as reduced OXPHOS protein levels for complexes II and III, were detected. An analysis of PINK1 and parkin, two proteins which are linked to parkin-mediated mitophagy and recruited to depolarized mitochondria, demonstrated dysfunctional recruitment of parkin, a known interaction partner of ATXN3, under CCCP treatment in SCA3 patient-derived fibroblasts. Furthermore, we were able to link the mitochondrial phenotype observed in SCA3 patient-derived fibroblasts to canonical mitophagy by the differential expression of p62, dysfunctional conversion of LC3-I to LC3-II and increased number of autophagosomes. For the first time, we showed that VDAC1, a gatekeeper for metabolites, ions and nucleotides in mitochondria, was directly deubiquitinated by ataxin-3 and that SCA3 patient-derived fibroblasts demonstrated its differential ubiquitination under CCCP treatment. The data were partly confirmed using a second fibroblast cell line from a transgenic SCA3 mouse which, in some aspects, presented an even stronger mitochondrial phenotype.

## 3. Discussion

Mitochondria are essential organelles that execute and coordinate various central processes such as survival, proliferation, metabolism and migration in the cell. Therefore, mitochondrial dysfunction severely affects cell fitness and contributes to neurodegenerative diseases, including PD and AD, and polyQ-diseases, including HD and SCA3 [1,2]. To maintain an intact mitochondrial network and, therefore, adequate cellular homeostasis, dysfunctional mitochondria were selectively removed by a process known as mitophagy. The best characterized mitophagy pathway is driven by the E3 ubiquitin ligase parkin and the PTEN-induced putative kinase 1 (PINK1), both linked to familial PD [26]. However, there is also growing evidence that mitophagy can be still functional in the absence of parkin by including other E3 ligases such as MUL1, SIAH1 or ARIH1 in this process [31]. For the autosomal dominantly inherited neurodegenerative disease spinocerebellar ataxia type 3 (SCA3), several studies investigated mitochondrial dysfunction in SCA3 cell and animal models [3,4,5,6,7]. It was shown that ataxin-3 itself can localize to mitochondria [19,20] and, therefore, may influence directly mitochondrial function under normal and disease conditions. Based on these findings, we have now characterized mitochondrial morphology, function and degradation of damaged mitochondria by parkin-mediated mitophagy in two different SCA3 fibroblast cell lines: one generated from an SCA3 patient skin biopsy, and the other from an embryonic skin biopsy of an SCA3 transgenic mouse model expressing polyQ-expanded ataxin-3 with 148Q [24].

An interesting approach to access the role of ataxin-3, the disease protein in SCA3, in mitochondrial dysfunction is to look at the pivotal role of ataxin-3 in cellular protein control systems by its function as a deubiquitinating enzyme (DUB). For instance, ataxin-3 preferentially cleaves mixed linked K48-K63 polyubiquitin chains from substrates [8]. One known substrate of ataxin-3 is the already mentioned protein parkin (encoded by *PARK2*) [9,10,11], which plays a key role in mitochondrial quality control. Parkin ubiquitinates mitochondrial outer membrane proteins such as VDAC1 (voltage-dependent anion channel 1) [30] and mitofusins, and thereby recruits autophagic proteins such as p62 to induce degradation of dysfunctional mitochondria by mitophagy [12]. To gain further understanding on how depolarized mitochondria, as also found here in our SCA3 cell models, are degraded in the context of expanded ataxin-3, we further analyzed canonical parkin-mediated mitophagy in our two cell models. Similar to our former published mouse study expressing an N-terminal ataxin-3 fragment [7], we demonstrated in this study that the ratio of LC3-II/LC3-I was similar in SCA3 patient fibroblasts and fibroblasts from healthy controls under normal growth conditions. Under depolarized conditions, the selective autophagy adaptor protein p62 and the ratio of LC3-II/LC3-I were increased in both and were overactivated in SCA3 patient fibroblasts compared to healthy controls. p62 and LC3 are known to interact with ubiquitinated mitochondrial proteins, which are labeled for mitophagy and, therefore, facilitate the recruitment of damaged mitochondria to autophagosomes [30,32]. Treatment of human neuroblastoma cell lines with the mitochondrial uncoupler CCCP revealed a co-localization of p62 with TOM20 and resulted in increased p62 protein levels in the cytoplasm. Additionally, increased LC3-II protein levels were observed following CCCP as well as bafilomycin treatment [33]. In line with that, Ivankovic et al. and we demonstrated higher autophagosome numbers after induction of autophagy by CCCP/rapamycin and inhibition of autophagosome degradation. These results point towards an impaired mitophagy in our SCA3 patient fibroblasts and are consistent with a study which analyzed the expression of autophagic proteins in post-mortem SCA3 brains [34].

In SCA3, metabolic dysfunction and body weight reduction are widely known in patients and animal models [35,36,37]. Recently, proteomics revealed a wide range of metabolic and mitochondrial dysregulated proteins in an SCA3 knockin mouse model in early pathogenesis. In line with the described protein dysregulation, these mice demonstrated reduced oxygen consumption rates and failed to gain weight, which points to an important defect in mitochondrial function and energy metabolism early in SCA3 pathogenesis [22]. Importantly, SCA3 patients also show a decreased BMI, which is inversely correlated with the expanded CAG repeat length [38] and could be a predictor of disease progression, as shown in a Chinese SCA3 cohort [39]. Body weight reduction and metabolic dysfunction are often observed in SCA3, including decreased glucose metabolism in the cerebellum, pons, and frontal cortex [40]. A gatekeeper for metabolites, as described above, but also for nucleotides and ions in mitochondria, is the voltage-dependent anion-selective channel 1 (VDAC1) that regulates mitochondrial function (reviewed in [41]) and is a very strong substrate of parkin-mediated polyubiquitination. When parkin induces polyubiquitination on VDAC1, the ubiquitinated VDAC1 triggers parkin-mediated mitophagy by recruiting p62/sequestosome-1 (SQSTM1) and LC3-II to mitochondria [30]. On the other hand, VDAC1 also regulates apoptosis by controlling the opening and closing of the mitochondrial permeability transition pore (mPTP). Interestingly, it was proposed that monoubiquitinated VDAC1 delays cell death by altering mitochondrial calcium levels, whereas polyubiquitinated VDAC1 plays a role in mitophagy induction [29]. In the present study, we analyzed if ataxin-3, a deubiquitinating enzyme, can directly influence mitochondrial dysfunction and mitophagy by VDAC1 deubiquitination. Interestingly, we demonstrated that mouse embryonic fibroblasts expressing ataxin-3 148Q have significantly higher levels of unmodified full-length and a higher ratio of polyubiquitinated form 2 of VDAC1, but significantly lower ratios of monoubiquitinated VDAC1 and the polyubiquitinated form 3, compared to wildtype controls. Upon induction of mitochondrial stress, we detected a significant increase in unmodified full-length VDAC1 in wildtype mouse embryonic fibroblasts, but not in fibroblast expressing polyQ-expanded ataxin-3 (MEF 148Q). Ratios of monoubiquitinated and polyubiquitinated VDAC1 to full-length VDAC1 protein levels remained similar in control fibroblasts and fibroblast expressing polyQ-expanded ataxin-3. Similar results were also obtained in SCA3 patient fibroblasts. Here, under depolarized conditions, the patient-derived fibroblasts failed to express more mono- and polyubiquitinated VDAC1 compared to fibroblasts from healthy controls. In a PD study, it was demonstrated that VDAC1 mono- and polyubiquitination plays an important role in the decision of mitophagy and apoptosis. Additionally, it was shown that the absence of VDAC1 polyubiquitination hinders the recruitment of parkin to the mitochondria and subsequently impairs mitophagy [29]. This is concurrent with our findings, where we observed a reduced polyubiquitination of VDAC1 and no recruitment of parkin to mitochondria under depolarized conditions in SCA3 patient-derived fibroblast cell lines.

The increase in unmodified full-length VDAC1 levels and the decrease in ubiquitinated VDAC1 levels in our SCA3 cell models point to a compromised VDAC1 (de)ubiquitination in SCA3. This impairment can be explained in two different ways: either polyQ-expanded ataxin-3 itself deubiquitinates VDAC1 more efficiently than its wildtype counterpart under non-pathological conditions or polyQ-expanded ataxin-3 indirectly influences VDAC1 ubiquitination levels by acting on its E3 ubiquitin ligase parkin. Both explanations follow findings of an earlier study which demonstrated that polyQ-expanded ataxin-3 is catalytically more active [10,42]. To further investigate if ataxin-3 can directly influence VDAC1 deubiquitination, we performed an in vitro deubiquitination assay. Here, we demonstrated for the first time that ataxin-3 can deubiquitinate low- and high-molecular polyubiquitinated VDAC1. This effect was even stronger in patient-derived fibroblasts. Therefore, our data point to VDAC1 as a new substrate of the deubiquitinase ataxin-3 and suggest that ataxin-3, a protein known to localize to mitochondria, is a crucial component of the canonical parkin-VDAC1-mediated mitophagy.

In summary, our SCA3 cell models demonstrated fragmented, circular mitochondria with reduced OXPHOS complexes, ATP production and cell viability (as illustrated in Figure 9). An important role in mitochondria quality control pathways is described for VDAC1 and its mono- and polyubiquitinated protein forms, which is a critical substrate of parkin responsible for the regulation of mitophagy and apoptosis. As ataxin-3 is functioning as a deubiquitinating enzyme and known to deubiquitinate parkin, we analyzed if ataxin-3 is an important component of the canonical parkin-mediated autophagy, which would explain the observed mitochondrial dysfunction and increased mitophagy. We demonstrated for the first time that ataxin-3 deubiquitinates VDAC1 directly and is leading to dysregulation of mono- and polyubiquitinated VDAC1 in SCA3 disease conditions. Consistently, we found reduced polyubiquitination of VDAC1 and no recruitment of parkin to mitochondria under depolarized conditions in SCA3 patient-derived cell lines, which points to a parkin-independent mitophagy or apoptotic pathway. For apoptotic pathways, several studies already demonstrated apoptotic processes at late disease stages in SCA3 [5,6,7,43]. However, for parkin-independent mitophagy further studies need to be performed, preferentially in cells not expressing parkin (like HeLa cells), to clarify which parkin-independent mitophagy pathways are linked to SCA3 pathogenesis.

## 4. Materials and Methods

### 4.1. Ethics Use of Animals

All mice were maintained by the animal care staff and veterinarians of the University of Tübingen. All procedures were performed according to the German Animal Welfare Act and the guidelines of the Federation of European Laboratory Animal Science Associations, based on European Union legislation (Directive 2010/63/EU). Animal experiments were approved by the local ethics committee (Regierungspräsidium Tübingen approved Anz. 17./18.12.2008).

### 4.2. Ethics Use of Human Tissue

All the work involving human tissue has been carried out in accordance with the Code of Ethics of the World Medical Association (Declaration of Helsinki) and with national legislation as well as our institutional guidelines. All experiments were approved by the Ethics committee of the University Hospital Tübingen, approval number 598/2011.

### 4.3. Generation of Mouse Embryonic Fibroblast Lines

Mouse embryonic fibroblast (MEF) lines were generated from mice expressing human full-length ataxin-3 with 148 glutamine repeats (MEF 148Q) under the control of the huntingtin promoter [24] and from wildtype littermates, as described previously [43].

### 4.4. Immortalization of Human Fibroblasts (iHF)

Human fibroblast cultures from an SCA3 patient (iHF SCA3: female, age 66, 23 and 67 CAG repeats in *MJD1*) and a sex- and age-matched healthy control (iHF hc: female, age 69, 14 and 21 CAG repeats in *MJD1*) were immortalized at passage four by transduction with the large T antigen of the simian virus 40 (SV40) (BioCat, Heidelberg, Germany). For transduction, Dulbecco’s Modified Eagle Medium (DMEM) GlutaMAX growth medium (Life Technologies, Carlsbad, CA, USA) was replaced by 1 mL of fresh medium containing 10 ng/mL of Polybrene (hexadimethrine bromide, Sigma Aldrich, Burlington, MA, USA), 1 mL of virus solution was added and cells were incubated with the virus for 24 h at 37 °C. After incubation, the medium was withdrawn, cells were washed with Dulbecco’s phosphate-buffered saline (PBS, Life Technologies, Carlsbad, CA, USA) and 2 mL of fresh medium without Polybrene was added. When the well was fully grown, cells were detached using 0.25% trypsin (Life Technologies, Carlsbad, CA, USA) and were transferred to a new cell culture dish (10 cm diameter). Cells were split 1:10 after growing to full confluence. Four passages post-transfection, the culture contained mainly immortalized cells, which was determined by comparing the culture to non-transfected control cell cultures. Cells were aliquoted and stored at −80 °C until further use.

### 4.5. Cell Culture Maintenance and Induction of Mitochondrial Stress and Autophagy

iHF and MEF were cultivated at 37 °C and 5% CO_2_ in DMEM—with (iHF) or without (MEF) GlutaMAX—and supplemented with 10% FBS and 100 U/mL of penicillin/streptomycin (both, Life Technologies, Carlsbad, CA, USA). Mitochondrial stress was induced by cultivating cells in growth medium containing 50 µM of carbonyl cyanide m-chlorophenyl hydrazone (CCCP, Biomol, Hamburg, Germany) for 6 h. Autophagy was induced by cultivating cells in growth medium containing 400 nM of rapamycin (Merck Millipore, Burlington, MA, USA) for 6 h. To induce autophagy and block the fusion of autophagosomes and lysosomes, cells were cultivated in growth medium containing 400 nM of rapamycin for 4 h, followed by a 2 h cultivation in fresh growth medium containing 400 nM of rapamycin and 50 nM of bafilomycin (Invivogen, San Diego, CA, USA). Controls were cultivated in growth medium containing dimethyl sulfide (DMSO, Roth, Germany), which was used as a solvent for rapamycin and bafilomycin for equivalent time periods.

### 4.6. Cell Viability Assay

Cell viability was determined using the resazurin-based PrestoBlue Cell Viability Reagent (Thermo Fisher Scientific, Waltham, MA, USA) according to the manufacturer’s protocol. For this, 20,000 cells were seeded in a 96-well plate (Greiner Bio-One, Frickenhausen, Germany) in 200 µL of medium and maintained under standard culturing conditions for 24 h. Afterwards, growth medium was replaced with 100 µL of fresh medium containing 10% PrestoBlue reagent and cells were incubated at 37 °C for 1 h. After incubation, fluorescence signals were measured using an Envision Multilabel Reader (PerkinElmer, Waltham, MA, USA) equipped with a FITC 535-nm emission and Cy5 620-nm excitation filters.

### 4.7. ATP Production Assay

Relative ATP levels were measured using the firefly luciferase-based ATPlite^TM^ Luminescence Assay System (PerkinElmer, Waltham, MA, USA) according to the manufacturer’s protocol. Cells were grown for 24 h in 96-well plates, the medium was replaced with 100 µL of fresh medium and 50 µL of cell lysis solution was added. Cells were incubated light protected at room temperature with gentle shaking for 5 min. To each well, 50 µL of substrate solution was added and cells were incubated light protected for another 15 min. Luminescence was measured using an Envision Multilabel Reader (PerkinElmer, Waltham, MA, USA).

### 4.8. Fluorescence-Activated Cell Sorting (FACS)

Cells were plated one day prior to measurements in 6-well plates with a density of 200,000 cells per well. The next day, cells were incubated in 250 µM of tetramethylrhodamine ethyl ester perchlorate (TMRE, Sigma-Aldrich, Burlington, MA, USA) for 30 min. For each genotype, one well of cells served as the positive control and was treated with 50 µM of CCCP for 6 h prior to staining. After TMRE incubation, cells were washed with PBS and detached from the plate using a cell scraper. Cells were collected in 1 mL of PBS and pelleted by centrifugation. The pellet was resuspended in 1 mL of fresh PBS and cells were transferred to a 5-mL FACS tube and stored light protected on ice until measurements were started. Using a CyAn ADP Analyzer (Beckman Coulter, Brea, CA, USA), cells were sorted according to their fluorescence. Therefore, the 635-nM filter was opened, 20,000 cells per sample were allowed to pass through and the number of fluorescent cells was counted by the Summit 4.3 software (Beckman Coulter, Brea, CA, USA).

### 4.9. Mitochondrial Labeling and Fluorescence Microscopy

iHF were grown on cover slips coated with poly-L-Lysine and transfected with pDsRed2-Mito plasmid (ClonTech, Mountain View, CA, USA) and pEGFP-LC3 plasmid (kindly provided by Y. Kabeya [44]) using Attractene (Qiagen, Hilden, Germany) according to the manufacturer’s protocol. Mitochondrial stress and autophagy were induced as described above and fixation was carried out 48 h after transfection using 4% PFA. For this, 200 µL of 4% PFA were added to the cells growing in a total volume of 2-mL medium, and cells were incubated at 37 °C for 10 min. In the second step, the medium was removed, and cells were incubated in pure 4% PFA at room temperature for 15 min under gentle shaking. Cells were washed with PBS three times for a total of 30 min and mounted on microscope slides using the VECTASHIELD^®^ mounting medium with DAPI (Vector Laboratories, Burlingame, CA, USA). Fluorescence microscopy was performed using a Zeiss microscopy system consisting of an Axiovert 200M microscope with apotome and the AxioVision software for fixed human fibroblasts (Zeiss, Jena, Germany). Images were analyzed using ImageJ software (Wayne Rasband, NIH) with a specialized mitochondrial morphology plug-in provided by Ruben Dagda [45].

In MEF, mitochondria were stained with MitoTracker^®^ Green FM (Life Technologies, Carlsbad, CA, USA) and nuclei with Hoechst 33342 (Thermo Scientific, Waltham, MA, USA) according to the manufacturer’s protocol. Cells were imaged with a live-cell fluorescence microscope (Axiovert Plan-Apochromat 63 × 1.4, Zeiss) at 37 °C and 5% CO_2_.

For both, experiments were repeated 3 times independently and the order in which wildtype cells (MEF et/wt or iHF hc) and SCA3 cells (MEF 148Q or iHF SCA3) were imaged were altered.

### 4.10. Electron Microscopy

Cells were fixed with 2.5% glutaraldehyde (Electron Microscopy Sciences, Carlsbad, CA, USA) and 4% PFA in sodium cacodylate buffer (0.1 M of sodium cacodylate, pH 7.4, Roth, Karlsruhe, Germany) overnight at 4 °C. The next day, the buffer was replaced with a sodium cacodylate buffer without glutaraldehyde and stored at 4 °C until further processing. Samples were stained with osmium tetroxide and dehydrated by passing through a series of washes with increasing ethanol concentration (50%, 70%, 80%, 90% and 100% Ethanol, 30 min for each concentration). Samples were incubated in propylene oxide/araldite overnight and then embedded in pure araldite. Semithin sections with a thickness of 0.4 µm were generated using a Ultracut R-Microtome (Leica Microsystems, Wetzlar, Germany) and stained with Richardson solution. Afterwards, ultrathin sections were cut with a thickness of 50 nm and stained with lead citrate and uranyl acetate. Ultrathin sections were examined using an EM 10-Electron microscope (Zeiss, Jena, Germany).

### 4.11. Western Blot Analysis

Western blot analysis was performed as described previously [46]. Protein concentration was determined using the Bio-Rad protein assay according to the manufacturer’s protocol and 30 µg of protein (for parkin, 50 µg) were loaded on Bis-Tris SDS polyacrylamide gels and separated electrophoretically. After electrophoresis, proteins were transferred onto a nitrocellulose membrane (GE Healthcare, Solingen, Germany). The transfer was carried out in a corresponding Bis-Tris/Bicine transfer buffer containing 15% methanol at 4 °C for 2 h at 80 V or overnight at 35 V. The membrane was incubated in a TBST buffer (10 mM Tris pH 7.5, 0.15 M NaCl, 0.1% (*v*/*v*) Tween-20) containing 5% milk powder (Naturaflor, Vechta, Germany) for 1 h and incubated with a primary antibody (see Table 1) diluted in TBST overnight at 4 °C. The membrane was washed and incubated for 2 h with a fluorescence- or horseradish peroxidase (HRP)-labeled secondary antibody at room temperature. In the case of a fluorescent labeled antibody, the signal was directly detected in the Odyssey FC Imager (LI-COR Biosciences, Bad Homburg vor der Höhe, Germany) under light with an appropriate wavelength. If the HRP-labeled antibody was used, membranes were incubated with ECL or ECL prime (GE Healthcare, Amersham, UK) for 1 min and a light signal was detected. For the Western blots shown in Figure 1, Figure 2, Figure 4, Figure 6 and Appendix A, protein bands were visualized using the enhanced chemiluminescence method (ECL, GE Healthcare, Amersham, UK) by exposure to Hyperfilm ECL (GE Healthcare, Amersham, UK). Obtained bands were quantified using the tools provided by the ImageStudio software (LI-COR Biosciences, Frankfurt, Germany) or ImageJ.

### 4.12. In Vitro Deubiquitination Assay

In vitro deubiquitination assays have been described earlier [47]. Briefly, proteins were extracted from immortalized human fibroblasts using a DUB assay buffer. Extracts were diluted in the same buffer to final protein concentration of 1.5 μg/μL and incubated with 0.25 μM of human His_6_-ataxin-3 (E-341, Bio-Techne, Minneapolis, MN, USA) at 37 °C for up to 2 h. Reactions were terminated with a 4× LDS sample buffer and 100 mM of dithiothreitol.

### 4.13. Statistical Analyses

Data are presented as mean and SEM. Datasets were tested for normality using the Shapiro–Wilk test. Nonparametric group analyses were performed using a two-sided Mann–Whitney *U* test, and using the Bonferroni correction for multiple comparisons. 

Statistical significance of the dataset obtained from Western blot analyses were determined using one-way ANOVA and Tukey’s post-test. For comparison of two different genotypes under normal growth conditions, an unpaired *t*-test was performed. To compare normal growth and treated conditions evaluating different genotypes, a one-way ANOVA was performed. For in vitro deubiquitination assays, curves were fitted to data points based on a one-phase nonlinear regression model. Statistical significance was demonstrated by *p*-values (≤0.05 (*), ≤0.01 (**), ≤0.001 (***)). All statistical and graphical evaluations were performed with GraphPad prism 8.0.

## Figures and Tables

**Figure 1 ijms-23-05933-f001:**
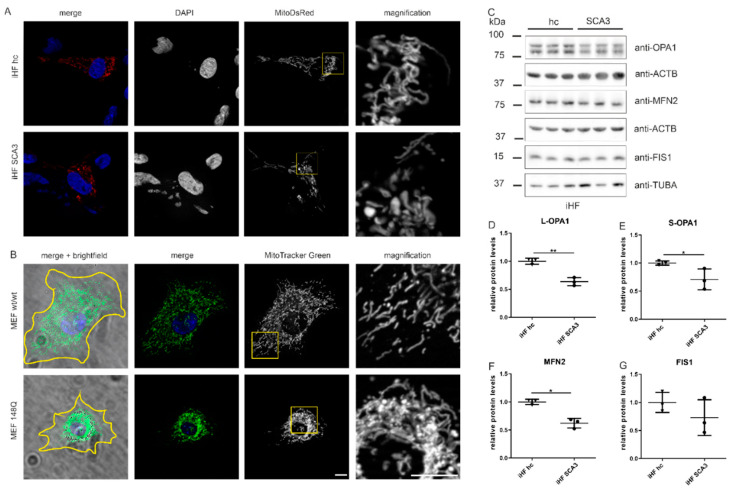
Mitochondrial morphology is impaired in SCA3 cell models linked to reduced fusion protein levels. (**A**) Fibroblast cultures from SCA3 patients (iHF SCA3) were transfected with MitoDsRed and fixed 48 h after transfection. Cells were mounted using VECTASHIELD mounting medium with DAPI for nuclear staining. Mitochondrial morphology was observed by fluorescence microscopy in three independent experiments. (**B**) Mouse embryonic fibroblasts derived from transgenic SCA3 mice (MEF 148Q) were stained with MitoTracker Green TM and nuclei were stained with Hoechst 33342. Cells were imaged with a life-cell fluorescence microscope at 37 °C and 5% CO_2_. The experiment was repeated three times and the order in which wildtype and MEF 148Q were imaged was altered. Yellow line in merged pictures indicates the cell body outline as observed by brightfield microscopy. (**A**,**B**) Scale bar indicates 10 µm and in higher magnification 20 µm. Yellow boxes indicate magnified regions. (**C**) Protein levels of OPA1, MFN2 and FIS1 were measured by Western blot analyses. ACTB or TUBA is shown as loading control. (**D**–**G**) Statistical analyses were determined using one-way ANOVA and Tukey’s post-test. iHF SCA3 were normalized to iHF hc. L(ong)-Opa1 = upper band, S(hort)-Opa1 = lower band. * *p* < 0.05, ** *p* < 0.01 by one-way ANOVA. Values are shown as mean +/− SEM. N = 3. hc = healthy control, SCA3 = fibroblasts derived from SCA3 patient, iHF = immortalized human fibroblasts, MEF = mouse embryonic fibroblasts, wt = wildtype.

**Figure 2 ijms-23-05933-f002:**
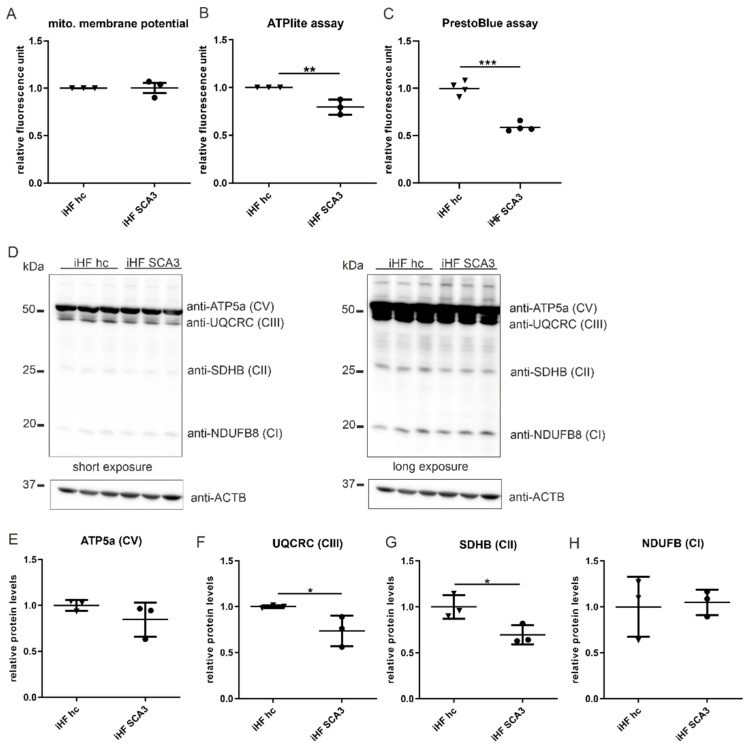
Impaired expression of OXPHOS proteins, ATP production and cell viability in SCA3 fibroblasts. (**A**) Mitochondrial membrane potential was investigated by TMRE staining, followed by fluorescence-activated cell sorting. The experiment was repeated three times independently and three samples per genotype were measured per experiment. Statistical analysis of the mean TMRE signal intensity per sample is demonstrated. (**B**) For luciferase-based ATP measurements using ATPlite assay, cells were seeded in 96-well plates 24 h prior to the experiment. Three independent experiments were performed. Luminescence was measured using Perkin Elmer reader. (**C**) Cell viability was determined in vitro using PrestoBlue Cell Viability assay. Therefore, cells were grown in 96-well plates for 24 h. Fluorescence was measured by an ELISA reader at 540/590 nm. (**D**) Protein levels of NDFUB, UQCRC2, SDHB and ATP5a were analyzed by Western blotting using an antibody mixture of different OXPHOS proteins. For better visualization of the different OXPHOS proteins, different exposures are shown. ß-actin (ACTB) is shown as loading control. (**E**–**H**) Statistical analyses were determined using one-way ANOVA and Tukey’s post-test. iHF SCA3 were normalized to iHF hc. * *p* < 0.05, ** *p* < 0.01, *** *p* < 0.001 by one-way ANOVA. Values are shown as mean +/− SEM. N = 3. hc = healthy control, SCA3 = fibroblasts derived from SCA3 patient, iHF = immortalized human fibroblasts.

**Figure 3 ijms-23-05933-f003:**
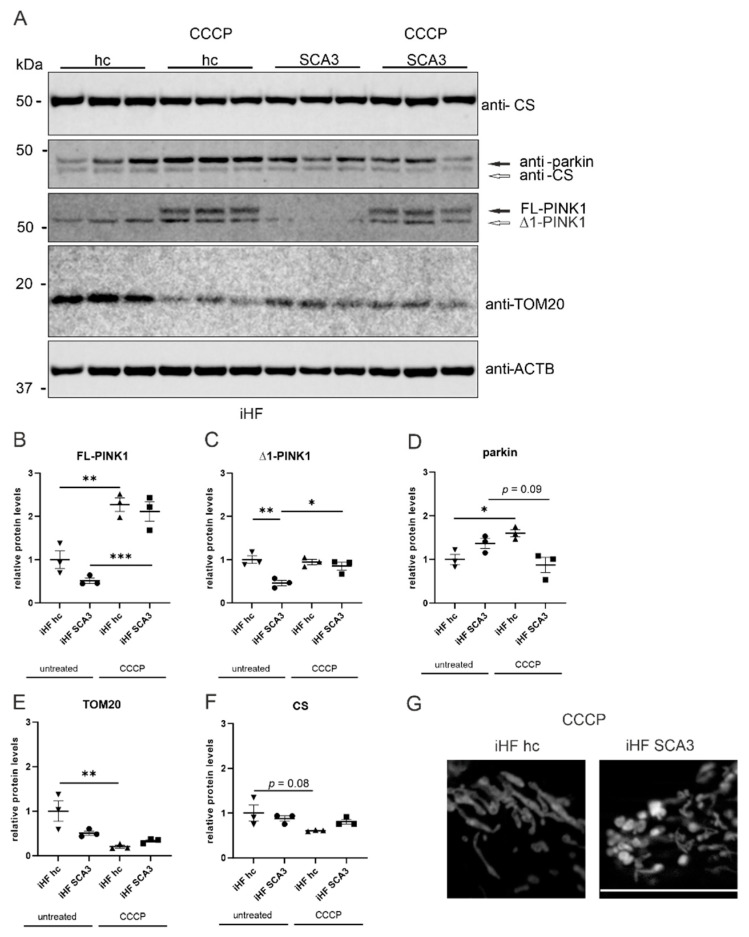
Parkin levels are not increased after CCCP treatment in SCA3 patient-derived fibroblasts. (**A**) Protein levels of parkin, PINK1, citrate synthase (CS) and TOM20 were measured in patient-derived fibroblasts. Beta-actin (ACTB) is shown as loading control. (**B**–**F**) Statistical analyses were performed from three independent experiments by one-way ANOVA and Tukey’s multiple comparison test. All genotypes and treatments were normalized to untreated iHF hc. (**G**) Fibroblasts were transfected with MitoDsRed vector and fixed 48 h after transfection. Depolarization of mitochondrial membrane potential was induced by 50 µM CCCP for 6h prior to fixation. Scale bar indicates 20 µm. * *p* < 0.05, ** *p* < 0.01, *** *p* < 0.001. Values are shown as mean +/− SEM. N = 3. hc = healthy control, SCA3 = fibroblasts derived from patients, iHF = immortalized human fibroblasts, FL= full-length, ∆1 = cleaved, CCCP = carbonyl cyanide m-chlorophenyl hydrazone.

**Figure 4 ijms-23-05933-f004:**
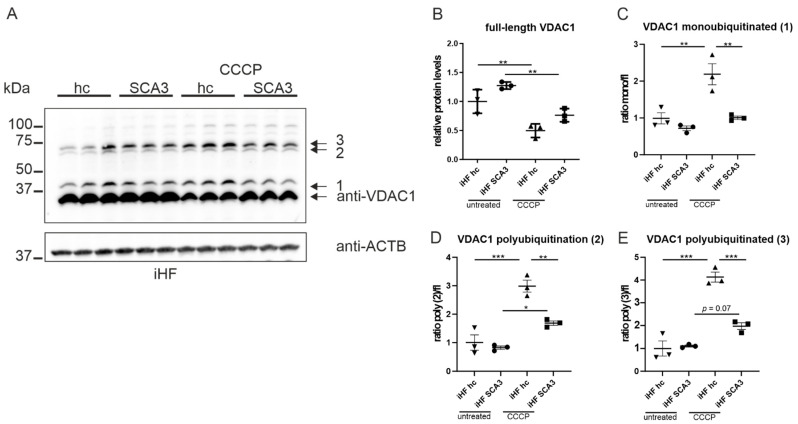
PolyQ-expanded ataxin-3 modulates VDAC1 ubiquitination. (**A**) Protein levels of full-length, mono- and polyubiquitinated VDAC1 under normal growth conditions as well as after 6 h of 50 µM CCCP treatment. ACTB is shown as loading control. (**B**–**E**) Ratio of ubiquitinated forms of VDAC1 was calculated to full-length expression values of VDAC1. All genotypes and treatments were normalized to untreated iHF hc. * *p* < 0.05, ** *p* < 0.01, *** *p* < 0.001 by one-way ANOVA. Values are shown as mean +/− SEM. N = 3.

**Figure 5 ijms-23-05933-f005:**
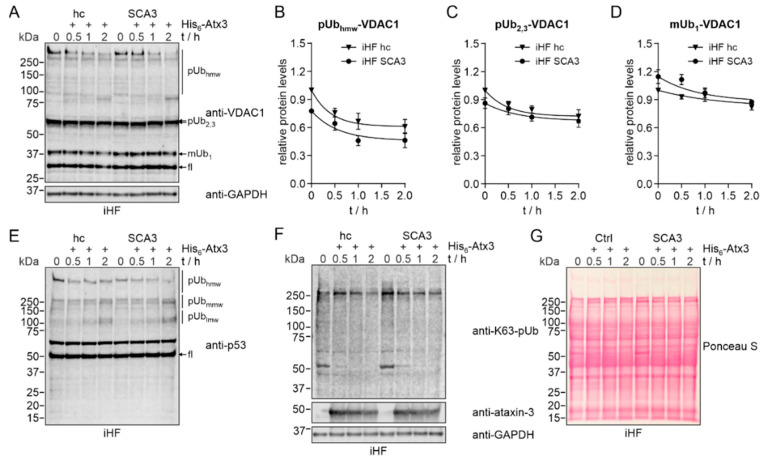
Ataxin-3 deubiquitinates VDAC1 in vitro. Cell extracts of immortalized human fibroblasts from healthy controls (iHF hc) or SCA3 patients (iHF SCA3) were incubated with purified His6-ataxin-3 for up to 2 h. (**A**) Western blot analysis of VDAC1 revealed a time-dependent reduction in monoubiquitinated (mUb1) forms of VDAC1, and low-molecular (2,3, lmw) and high-molecular weight (hmw) polyubiquitinated (pUb) forms of VDAC1. GAPDH served as a loading control. fl = full-length VDAC1. (**B**–**D**) For quantitative analysis, curves were extrapolated based on a one-phase decay nonlinear regression model. N = 3. (**E**,**F**) As assay controls, ataxin-3-mediated breakdown of K63-linked polyubiquitin chains and of lmw, medium-molecular (mmw) and hmw pUb forms of p53 were detected. Addition of His6-ataxin-3 was confirmed by immunodetection. GAPDH served as loading control. (**G**) Total protein staining with Ponceau S was performed as an additional loading control and for monitoring protein integrity.

**Figure 6 ijms-23-05933-f006:**
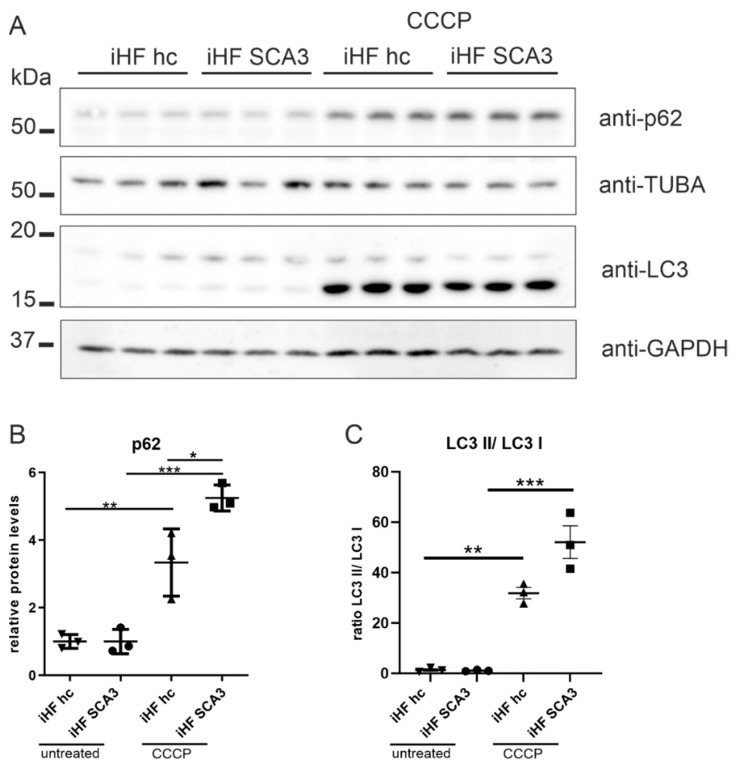
Autophagic markers LC3 and p62 are altered in SCA3 patient-derived fibroblasts. (**A**) Protein levels of p62 and LC3 were analyzed by Western blot analyses under normal growth conditions and after 6 h CCCP treatment (50 µm). GAPDH or TUBA was used as loading control. (**B**,**C**) Statistical analyses were performed by one-way ANOVA and Tukey’s post-test. All genotypes and treatments were normalized to untreated iHF hc. * *p* < 0.05, ** *p* < 0.01, *** *p* < 0.001 by one-way ANOVA. Values are shown as mean +/− SEM. N = 3. hc = healthy control, SCA3 = fibroblasts derived from patients, iHF = immortalized human fibroblasts.

**Figure 7 ijms-23-05933-f007:**
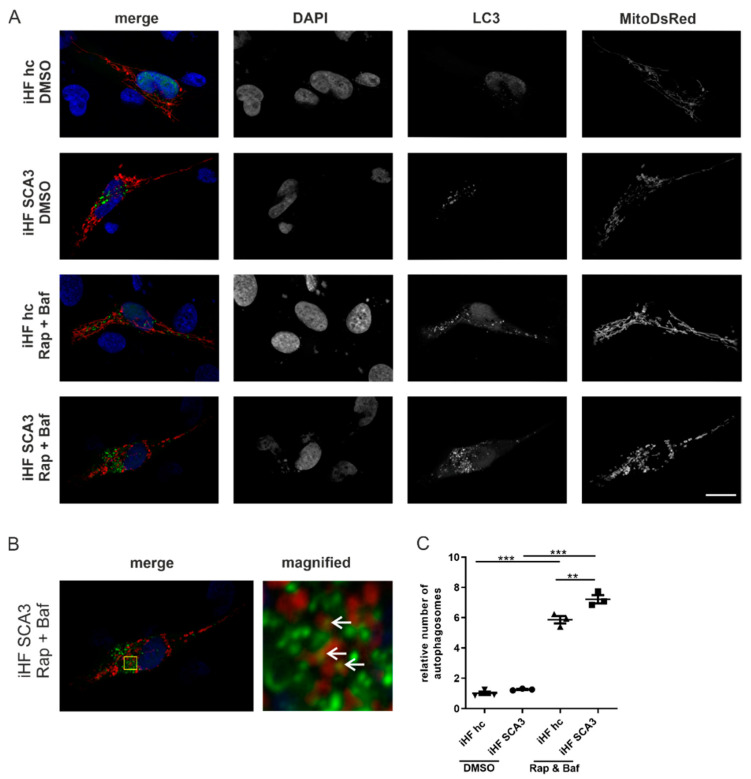
Increased number of autophagosomes in SCA3 patient-derived fibroblasts. SCA3 patient and control fibroblasts were co-transfected with pEGFP-LC3 and pDsRed2-Mito. (**A**) iHF hc and iHF SCA3 were co-transfected with pEGFP-C1 LC3 and pDsRed2-Mito plasmids. Forty-two hours after transfection, fibroblasts were treated with rapamycin (rap; 400 nM, 6 h) and bafilomycin (baf; 50 nM, 2 h). After treatments, cells were fixed with PFA, nuclei stained with DAPI and images taken with an Axiovert 200 M microscope. (**B**) Autophagosomes (green) were in close proximity to mitochondria and partially overlapping with mitochondria (yellow area indicated by arrows). Yellow box indicates magnified region. (**C**) Number of autophagosomes was evaluated in approximately 30 cells from three independent experiments using ImageJ. ** *p* < 0.01, *** *p* < 0.001 by one-way ANOVA. Values are shown as mean +/− SEM. N = 3. hc = healthy control, SCA3 = fibroblasts derived from patients, iHF = immortalized human fibroblasts.

**Figure 8 ijms-23-05933-f008:**
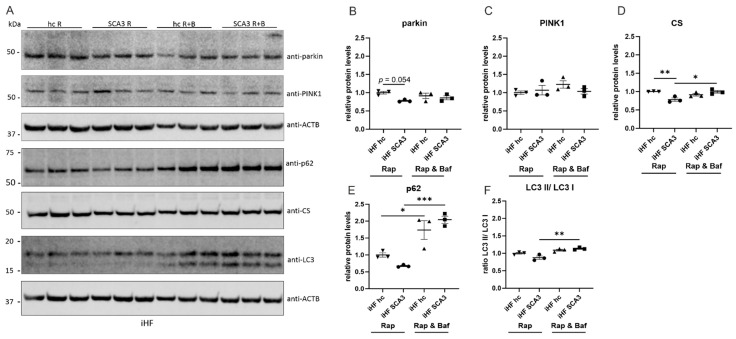
Autophagic markers LC3 and p62 are altered in SCA3 patient-derived fibroblasts. (**A**) Protein levels of parkin, PINK1, citrate synthase (CS), p62 and LC3 in iHF cell lines were analyzed by Western blot under normal growth conditions and after rapamycin (R, Rap, 400 nM, 6 h) and rapamycin/ bafilomycin (B, Baf, 50 nM, 2h and R, Rap, 400 nM, 6 h) treatments. ACTB was used as loading control. (**B**–**F**) Statistical analyses were performed using one-way ANOVA and Tukey´s multiple comparison test. The ratio of LC3-II/LC3-I was calculated by dividing the expression value of LC3-II by LC3-I. All genotypes and treatments were normalized to iHF hc. * *p* < 0.05, ** *p* < 0.01, *** *p* < 0.001 by one-way ANOVA. Values are shown as mean +/− SEM. N = 3. hc = healthy control, SCA3 = fibroblasts derived from patients, iHF = immortalized human fibroblasts.

**Figure 9 ijms-23-05933-f009:**
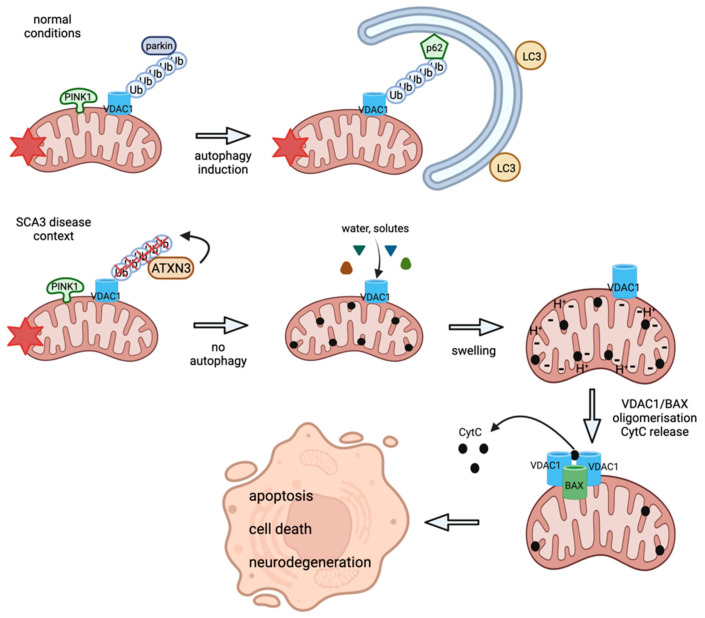
Proposed model of mitochondrial and mitophagic dysfunction in SCA3 cell models. Under normal conditions, autophosphorylated PINK1 accumulates at the OMM and recruits cytosolic parkin. Parkin ubiquitinates OMM-located proteins such as VDAC1 and labels them, and thereby the entire mitochondrion, for degradation by the proteasome or mitophagy. In an SCA3 disease context, mutant ATXN3 itself deubiquitinates VDAC1, which leads to reduced polyubiquitination of VDAC1 that hinders mitophagy and initiates apoptotic pathways by mitochondrial swelling, BAX/VDAC1 oligomerization and cytochrome c (CytC) release. In the end, apoptotic processes lead to cell death and neurodegeneration. Graph created with BioRender.com (accessed on 9 March 2022).

**Table 1 ijms-23-05933-t001:** Used antibodies in the present study.

Target Protein	Product Number	Species	Manufacturer	Dilution
ACTB	clone AC-15, A5441	mouse	Merck, Kenilworth, NJ, USA	1:10,000/ 1:5000
ATXN3 ATXN3	clone 13H9L9, 702788 clone 1H9, MAB5360	rabbit mouse	Thermo Fisher Scientific, Waltham, MA, USA Merck, Kenilworth, NJ, USA	1:1000 1:4000
CS FIS1	ab96600 ab96764	rabbit rabbit	Abcam, Cambridge, UK Abcam, Cambridge, UK	1:1000 1:2000
GAPDH	ab125247	mouse	Abcam, Cambridge, UK	1:2000
K63-pUb	clone D7A11, #5621	rabbit	Cell Signaling Technology, Danvers, MA, USA	1:500
LC3	0231	mouse	nanoTools, Freiburg, Germany	1:100
MFN2	ab104632	rabbit	Abcam, Cambridge, UK	1:2000
OPA1	612606	mouse	BD Bioscience, Franklin Lakes, NJ, USA	1:1000
OXPHOS	ab110413	mouse	Abcam, Cambridge, UK	1.1000
p53	clone 6H5E7, CSB-MA0240771A0m	mouse	Cusabio, Houston, TX, USA	1:500
p62	5114S	rabbit	Cell Signaling Technology, Danvers, MA, USA	1:1000
parkin PINK1 TOM20 TUBA	ab15954 sc-517353 ab56783 CP06	rabbit mouse mouse mouse	Abcam, Cambridge, UK Santa Cruz, Inc, Dallas, TX, USA Abcam, Cambridge, UK Merck, Kenilworth, NJ, USA	1:200 1:500 1:5000 1:5000
VDAC1	Ab10527	rabbit	Merck, Kenilworth, NJ, USA	1:10,000

## Data Availability

All data generated or analyzed during this study are included in this published article and its Appendix A.

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
