# Peer review of "Mitochondrial Dysfunction in Spinocerebellar Ataxia Type 3 Is Linked to VDAC1 Deubiquitination"

_ijms, 2022, doi:10.3390/ijms23115933_

Round 1
Reviewer 1 Report
Review ijms-1683154
Harmuth et al. present an interesting manuscript about the role of ataxin-3 in mitochondrial viability. They confirm that mitochondria from SCA3 patient fibroblasts are compromised in shape and function and show that VDAC1 ubiquitination patterns are different in mutant, polyQ ataxin-3 expressing cells.
The study is important for better understanding of SCA3 and provides some novel ideas about the connection ataxin-3/VDAC1, but needs some substantial revision before acceptable for publication. Main hypotheses and conclusions need to be stated clearly and discussed in regard to the obtained results.
major comments
- A general remark to the figure legends
I would find it easier to read the manuscript if the figure legends would contain more information about the respective experiments, including incubation times and concentration of compounds. Currently the legends only repeat what is already stated in the corresponding text.
- Fig. 2 (and all figures with Western Blots).
Are the blots shown really all from the same gel/membrane? Looks like the molecular weight of the proteins of interest is quite close to each other. If they are not all from the same membrane, it is misleading to show only on loading control and this has to be indicated in the figure legend.
- paragraph 2.3
“Parkin is not recruited…” is an overstatement. The western blots are obtained from total cell lysates (unless I missed something) and only allow conclusions about total protein levels, not recruitments or localizations.
- Fig. 3
These blots definitely can not come from one membrane, therefore it has to be clarified what the loading control shows. Please specify in the methods how the quantification was done, were the bands related to the loading control (which one)? Were the membranes reblotted etc. I recommend, also for the other figures, a supplemental figure with all full-size blots/membranes (rearrange the stripes, if membranes were cut).
- Fig. 4
The details of the quantification are not clear. Did you set the mean protein levels in iHF hc cells to 1, and the other values were related to that? Wouldn´t ratios be more informative, always related to the full-length VDAC1? What is the conclusion of this part?
- Fig. 6
Why are the p62 level increased after depolarization, both in hc and SCA3 cells? Upon stimulation of autophagy p62 levels normally decrease, and depolarization should stimulate mitophagy? Also the increased LC3-II levels indicate a stimulated autophagy?
- Fig. 8
The quantification graphs are not consistent and confusing. Firstly, for PINK1, were the mean relative protein levels of iHF hc cells treated with Rap set to 1? And the values of the other conditions related to that? For p62, CS, LC3 I, were the iHF SCA3 levels set to 1? Anyway, I think ratios are more informative. For example LC3 I/LC3 II in Rap-treated cells is X, in Rap&Baf -treated cells Y. p62 in iHF hc is x-fold higher upon rap+baf compared to rap only. etc….
- In the summary of the result section and in the discussion you avoid clear statements about what you think is going on. In German the expression is “Schwurbelei”. For example line 423 ff. What is the conclusion, do you want to say that in SCA3 mitophagy is increased in steady state, but decreased when mitos are depolarized? Another example, line 447 “Interestingly, we…” What is the conclusion? Are the cells then protected from apoptosis?
minor comment
-the figures are labeled A,B,C in capital letters, whereas in the corresponding legends a,b,c is used.
Author Response
Dear Editor,
Please find enclosed our revised manuscript „ Mitochondrial dysfunction in Spinocerebellar ataxia type 3 is linked to VDAC1 deubiquitination“ by Harmuth et al., which we had submitted under the reference ijms-1683154. We were very grateful for the reviewer assessments and constructive criticism.
We are very grateful to get the opportunity to submit a revised version of our manuscript to you, in which we addressed the remaining concerns and issues raised by the reviewers Our responses to the reviewer comments are enclosed on the following pages.
Hopefully, we could improve our manuscript to your satisfaction and our thorough revision will allow for a positive decision on publishing our work in International Journal of Molecular Sciences.
We are looking forward to hearing from you at your earliest convenience.
Yours sincerely,
Dr. Jeannette Hübener-Schmid
Institute of Medical Genetics and Applied Genomics
University of Tuebingen
Tuebingen, Germany
Reviewer 1:
Reviewer comment: Harmuth et al. present an interesting manuscript about the role of ataxin-3 in mitochondrial viability. They confirm that mitochondria from SCA3 patient fibroblasts are compromised in shape and function and show that VDAC1 ubiquitination patterns are different in mutant, polyQ ataxin-3 expressing cells. The study is important for better understanding of SCA3 and provides some novel ideas about the connection ataxin-3/VDAC1, but needs some substantial revision before acceptable for publication. Main hypotheses and conclusions need to be stated clearly and discussed in regard to the obtained results.
Comment to the reviewer: We thanks the reviewer for the kind words and the help improving the manuscript.
major comments
Reviewer comment: A general remark to the figure legends
I would find it easier to read the manuscript if the figure legends would contain more information about the respective experiments, including incubation times and concentration of compounds. Currently the legends only repeat what is already stated in the corresponding text.
Comment to the reviewer: We changed all figure legends accordingly and now included in the figure legends information about the experimental design and shortened the results within the figure legend to not repeat information.
Reviewer comment: Fig. 2 (and all figures with Western Blots). Are the blots shown really all from the same gel/membrane? Looks like the molecular weight of the proteins of interest is quite close to each other. If they are not all from the same membrane, it is misleading to show only on loading control and this has to be indicated in the figure legend.
Comment to the reviewer: The western blots shown in figure 2 are all from the same membrane incubated with an antibody mixture for OXPHOS proteins. This mixture contains different antibodies for OXPHOS proteins in one vial and therefore, they are detected at the same timepoint. As all proteins detected by that antibody mixture showed different expression values, we showed in figure 2D different exposure times of the same blot. We now indicated that in the figure legend and a full-size blot is shown in an additional document submitted with the revision. Additionally, for all western blots shown in that manuscript we included full-size western blots in that additional document.
Reviewer comment: paragraph 2.3
“Parkin is not recruited…” is an overstatement. The western blots are obtained from total cell lysates (unless I missed something) and only allow conclusions about total protein levels, not recruitments or localizations.
Comment to the reviewer: We agree with the reviewer on that and now writing about parkin expression values to avoid overstatements.
Reviewer comment: Fig. 3
These blots definitely can not come from one membrane, therefore it has to be clarified what the loading control shows. Please specify in the methods how the quantification was done, were the bands related to the loading control (which one)? Were the membranes reblotted etc. I recommend, also for the other figures, a supplemental figure with all full-size blots/membranes (rearrange the stripes, if membranes were cut).
Comment to the reviewer: All blots are from the same membrane and therefore, only one housekeeping protein is shown. For TOM20 the lower part of the membrane where cutted of before the first detection. The upper part were incubated twice with anti-rabbit antibodies and also twice with anti-mouse antibodies. Therefore, a redetection of anti-CS is visible in the detection of parkin (as indicated in the blot) and a redetection of beta-actin in PINK1 detection. All raw data are now included in an additional document.
Reviewer comment: Fig. 4
The details of the quantification are not clear. Did you set the mean protein levels in iHF hc cells to 1, and the other values were related to that? Wouldn´t ratios be more informative, always related to the full-length VDAC1? What is the conclusion of this part?
Comment to the reviewer: We now checked all quantifications shown in the manuscript, to avoid misunderstanding. Now all untreated wildtype cells (MEF wt/wt or iHF hc) were set at 1 and the SCA3 genotypes and treatments were normalized to them. Whenever possible, we calculated the ratios of full-length protein levels to ubiquitinated levels (VDAC1) or cleaved protein levels (LC3).
Reviewer comment: Fig. 6
Why are the p62 level increased after depolarization, both in hc and SCA3 cells? Upon stimulation of autophagy p62 levels normally decrease, and depolarization should stimulate mitophagy? Also the increased LC3-II levels indicate a stimulated autophagy?
Comment to the reviewer: After CCCP treatment it was shown earlier that p62 and LC3-II levels get increased in human neuroblastoma cell lines. We now included an additional discussion to that in the discussion section.
Reviewer comment: Fig. 8
The quantification graphs are not consistent and confusing. Firstly, for PINK1, were the mean relative protein levels of iHF hc cells treated with Rap set to 1? And the values of the other conditions related to that? For p62, CS, LC3 I, were the iHF SCA3 levels set to 1? Anyway, I think ratios are more informative. For example LC3 I/LC3 II in Rap-treated cells is X, in Rap&Baf -treated cells Y. p62 in iHF hc is x-fold higher upon rap+baf compared to rap only. etc….
Comment to the reviewer: We thanks the reviewer for that hind. As mentioned by the reviewer all quantifications are performed to iHF hc Rap treated. Therefore, we set all iHF hc Rap to one and quantified the other cells and treatments to it. This was changed now in the graph. We also agree with the reviewer that the LC3-II/LC3-I ratio is more informative, therefore, we now included that statistical analyses in figure 8F, but also in figure 6.
Reviewer comment: In the summary of the result section and in the discussion you avoid clear statements about what you think is going on. In German the expression is “Schwurbelei”. For example line 423 ff. What is the conclusion, do you want to say that in SCA3 mitophagy is increased in steady state, but decreased when mitos are depolarized? Another example, line 447 “Interestingly, we…” What is the conclusion? Are the cells then protected from apoptosis?
Comment to the reviewer: We went over the result and discussion section and hope that we now indicate our results more clear.
minor comment
Reviewer comment: the figures are labeled A,B,C in capital letters, whereas in the corresponding legends a,b,c is used.
Comment to the reviewer: We now labeled the figure legend accordingly with capital letters.
Reviewer 2 Report
The present work titled “Mitochondrial dysfunction in Spinocerebellar ataxia type 3 is 2 linked to VDAC1 deubiquitination” is an interesting approach using patients-derived SCA3 cell models to study mitochondrial dysfunction and mitophagy linked to the disease. The authors showed that their experimental SCA3 cell models have morphologically altered mitochondria with reduced mitochondrial activity. However, this work presents some issues that must be solved by the authors.
Major comments:
General:
- Authors must quantify mitochondrial morphology changes, as they did for autophagosomes counting.
- Statistical analysis must be better described: Method for multiple comparisons, tested for normality…
- Authors should reunify figures. Some figures can be unified as they are the same theme or part of the same result.
- Figures are not well marked; every image/graph should have its own letter in order to guide the reader to follow the explanation (i.e. Fig 1a, b, c….). Also, inset images should show where they are taken from (Fig. 7B).
Fig.1:
- Authors must explain why they have used MitoDsRed vector in human fibroblast and MTGreen in mouse MEfs.
- The use of Anti-TUBA marker should be explained, and also, why its expression is changed, which makes this marker unsuitable as housekeeping protein. Please, specify the housekeeping protein used.
- Authors must give an explanation about short and long OPA1 isoforms, as they could act in an opposite way in mitochondrial dynamics.
- The cell body outline should be observed by brightfield microscopy without necessity of yellow mark. Also, it should be marked in both type of cells.
- Measuring mitochondrial dynamic protein expression should be have been done in both type of SCA3 cell models.
Fig. 2:
- Please check labels in the different graphs. Unify criteria: or fluorescence, or RFU, or TMRE signal…
Fig. 3:
- Authors must give an explanation about the use of 2 CS WB bands.
- Please, follow a logical order when describing the figure (before B than C).
Fig. 4:
- VDAC1 ubiquitination detection experiment is not well explained. Should it be done by IP with anti-Ub / anti-VDAC antibodies? Please, check this experimental approach.
Fig. 5:
- This fig. is especially difficult to follow:
o Time in hours in A and in min. in B
o Missing VDAC full lenght data
o No graphs for p53 nor K63 data
o Which is the meaning of MJD in ponceau? Ponceau membrane is referred to which of the blots?
o Addition of ataxin 3 is confirmed only for one blot.
Fig. 7:
- Why colours in single channels photography are missing?
Fig. 8:
- Please, explain the biological meaning of differences observed in LC3 I and LC3 II expression with the experimental conditions.
Minor comments:
- Line 114: Mitofusion2 instead of Mitofusin2
- Results paragraph 2.4 is repeated.
- Check figure legend 5 format.
Author Response
Dear Editor,
Please find enclosed our revised manuscript „ Mitochondrial dysfunction in Spinocerebellar ataxia type 3 is linked to VDAC1 deubiquitination“ by Harmuth et al., which we had submitted under the reference ijms-1683154. We were very grateful for the reviewer assessments and constructive criticism.
We are very grateful to get the opportunity to submit a revised version of our manuscript to you, in which we addressed the remaining concerns and issues raised by the reviewers Our responses to the reviewer comments are enclosed on the following pages.
Hopefully, we could improve our manuscript to your satisfaction and our thorough revision will allow for a positive decision on publishing our work in International Journal of Molecular Sciences.
We are looking forward to hearing from you at your earliest convenience.
Yours sincerely,
Dr. Jeannette Hübener-Schmid
Institute of Medical Genetics and Applied Genomics
University of Tuebingen
Tuebingen, Germany
Reviewer 2:
Comments and Suggestions for Authors
Reviewer comment: The present work titled “Mitochondrial dysfunction in Spinocerebellar ataxia type 3 is 2 linked to VDAC1 deubiquitination” is an interesting approach using patients-derived SCA3 cell models to study mitochondrial dysfunction and mitophagy linked to the disease. The authors showed that their experimental SCA3 cell models have morphologically altered mitochondria with reduced mitochondrial activity. However, this work presents some issues that must be solved by the authors.
Comment to the reviewer: We thanks the reviewer for the time and good hints which helped to improve the manuscript.
Major comments: General:
Reviewer comment: Authors must quantify mitochondrial morphology changes, as they did for autophagosomes counting.
Comment to the reviewer: In our earlier studies we always quantified mitochondrial morphological changes using an Image J plugin published by Ruben Dagda (Dagda et al., 2009). But in this manuscript a quantification of the described morphological changes were not possible by that Image plugin. In MEF 148Q cells, the mitochondria clustered in 80-90% of all stained cells with intact nucleus around the nucleus. Because of the accumulation of mitochondria around the nucleus and their spatial condensation, it was not possible to evaluate the mitochondrial morphology with Image J. The software was not able to discriminate single mitochondria and counted several mitochondria in close proximity as one large mitochondrion. We now included an explanation for that in the result section.
Reviewer comment: Statistical analysis must be better described: Method for multiple comparisons, tested for normality…
Comment to the reviewer: We now included more information about statistical analyses performed in the manuscript in the Material and Method section, as well as wrote more details about experimental design and statistical analyses in the respective figure legends as also indicated by reviewer 1.
Reviewer comment: Authors should reunify figures. Some figures can be unified as they are the same theme or part of the same result.
Comment to the reviewer: We did not changed the number of figures or reunified figures together, because all our figures include already several western blots, immunofluorescence pictures and quantifications.
Reviewer comment: Figures are not well marked; every image/graph should have its own letter in order to guide the reader to follow the explanation (i.e. Fig 1a, b, c….). Also, inset images should show where they are taken from (Fig. 7B).
Comment to the reviewer: We thanks the reviewer for that hind and now marked each image/graph by its own letter. We included a yellow box in the merged picture to label the position where magnified picture was taken from and explained the meaning of the yellow box in the figure legend.
Reviewer comment: Fig.1: Authors must explain why they have used MitoDsRed vector in human fibroblast and MTGreen in mouse MEFs.
Comment to the reviewer: At the beginning of the study we established the immunofluorescence conditions to evaluate mitochondrial morphology. We found out, that in live cell microscopy the MTGreen gave us the best possibility to evaluate mitochondrial morphology. But in fixed cells MTGreen was not function well. Therefore, we decide there to use a transfection with a MitoDsRed vector. We now included more information in the figure legend and material and method section.
Reviewer comment: Fig.1: The use of Anti-TUBA marker should be explained, and also, why its expression is changed, which makes this marker unsuitable as housekeeping protein. Please, specify the housekeeping protein used.
Comment to the reviewer: As indicated in figure 1, we used ACTB and TUBA as loading control. For FIS1 detection we also tried ACTB as loading control, but had several air bubbles around that protein size. Therefore, we decide to use another housekeeping protein at a different size. ACTB, GAPDH and TUBA are often used housekeeping proteins in the literature. We agree with the reviewer that the loading of the blot is not optimal, but as not strong effect was visible, we decided not to repeat that western blot.
Reviewer comment: Fig.1: Authors must give an explanation about short and long OPA1 isoforms, as they could act in an opposite way in mitochondrial dynamics.
Comment to the reviewer: We included now more information about short and long OPA1 in that section.
Reviewer comment: Fig.1: The cell body outline should be observed by brightfield microscopy without necessity of yellow mark. Also, it should be marked in both type of cells.
Comment to the reviewer: We checked all our results from live cell microcopy again and were thinking that a presentation without the yellow line in MEF148Q do not demonstrate the cell size in a good manner to understand and see that the mitochondria are clustering more around the nucleus in MEF148Q. Also we checked the brightfield in the iHF cells during microscopy but as a normal distribution of mitochondria were observed in that cell type, we did not further visualize brightfield in that cell type.
Reviewer comment: Fig.1: Measuring mitochondrial dynamic protein expression should have been done in both type of SCA3 cell models.
Comment to the reviewer: We agree with the reviewer that this would be important. Nevertheless, in the short time frame of the revision (10 days) we were unable to perform that experiment. For further studies, we need to follow up on that.
Reviewer comment: Fig. 2: Please check labels in the different graphs. Unify criteria: or fluorescence, or RFU, or TMRE signal…
Comment to the reviewer: We unified the labels of the graphs in figure 2 and also checked all other figures that, whenever possible, the same labels are used.
Reviewer comment: Fig. 3: Authors must give an explanation about the use of 2 CS WB bands. Please, follow a logical order when describing the figure (before B than C).
Comment to the reviewer: The second CS band on the parkin Western blot is a redetection of the former CS detection. Both, CS and parkin, are generated in rabbit and therefore, a secondary rabbit antibody was used. We now included an additional document with all full-size western blots to avoid misunderstanding.
Reviewer comment: Fig. 4: VDAC1 ubiquitination detection experiment is not well explained. Should it be done by IP with anti-Ub / anti-VDAC antibodies? Please, check this experimental approach.
Comment to the reviewer: We included more experimental information in the figure legend and calculated the ratio of ubiquitinated VDAC forms to the full-length protein levels to be more clear in the description of the data.
Reviewer comment: Fig. 5: This fig. is especially difficult to follow:
o Time in hours in A and in min. in B
o Missing VDAC full length data
o No graphs for p53 nor K63 data
o Which is the meaning of MJD in ponceau? Ponceau membrane is referred to which of the blots?
o Addition of ataxin-3 is confirmed only for one blot.
Comment to the reviewer: To unify the figure and make it easier to follow the experiments we changed, as indicated by reviewer 2, the time to hours in all blots and changed MJD to SCA3 in the graph of ponceau staining. Additionally, we quantified full length VDAC and p53 as well as polyubiquitinated p53 and K63-linked polyubiqutin. As this control experiment were planned as qualitative analyses and the polyubiquitination for p53 and K63-linked ubiquitin chains is already known and published we performed that experiment only once. Nevertheless, the quantification is now shown in Supplementary Figure 4. Additionally, in the figure legend more information about the experimental design is demonstrated.
Reviewer comment: Fig. 7: Why colours in single channels photography are missing?
Comment to the reviewer: To better visualize the results we used black and white color of the single images and showed only the overlay picture in all colors. This is often done also in other publications. Additionally, it will also help people with red-green blindness to see and interpret the results.
Reviewer comment: Fig. 8: Please, explain the biological meaning of differences observed in LC3 I and LC3 II expression with the experimental conditions.
Comment to the reviewer: As also indicated by reviewer 1, we now demonstrate the ratio of LC3-II/LC3-I and explained further the relevance of p62 and LC3 in mitophagy in the discussion section.
Minor comments:
Reviewer comment: Line 114: Mitofusion2 instead of Mitofusin2
Comment to the reviewer: The typing error was changed accordingly.
Reviewer comment: Results paragraph 2.4 is repeated.
Comment to the reviewer: We changed the paragraph label to 2.5
Reviewer comment: Check figure legend 5 format.
Comment to the reviewer: We unified the format of legend 5 to the other figure legends.
Round 2
Reviewer 1 Report
The authors present a revised version of their manuscript dealing with mitochondrial dysfunction in SCA3. While I´m satisfied with all points regarding figure legends, quantification and description and discussion, I´m not at all convinced about the Western Blots.
- The additional document does not give more information and merely shows the figures in the text a bit enlarged. What I asked for was convincing evidence that all blots are from the same membrane. This could be done for example by rearranging the cut pieces of the membrane and take a picture of them, along with showing exposures of the Western Blots where the boundaries of the membrane pieces are visible. And, if the images of the Westerns were further cropped, by showing the originals.
- I think you try to cheat here and I frankly don´t believe the blots are from the same membrane/gel:
2a: Fig. 1. Anti-ACTB and anti-TUB both run around 37 kDa. But whereas TUBA runs almost exactly horizontal, ACTB has a slight "smiley" on the left part of the gel. How can that be?
2b: Fig. 3: If you compare the running behaviour of CS in the top Blot and, allegedly the very same Blot and membrane below, where additionally parkin was detected, they can not be the same! Both are not horizontal, but tilted in opposite directions!
2c: Fig. 6: p62 is horizontal, TUBA has a clear smiley, left part running faster that right part of the gel. This cannot be since the two proteins have such a similar molecular weight and run on similar positions in the gel.
Author Response
Comment to Reviewer 1:
We thank again Reviewer 1 for the time to review our manuscript carefully. For sure it is not our intention to demonstrate data where the reviewer but also the later readers are not convinced about our data and presentation of data.
In figure 1, an additional ACTB loading control is now included for OPA1. We are grateful that the reviewer has comment on this again. Therefore, we carefully rechecked our data and found out that the earlier presented loading control ACTB was only the respective loading control for MFN2.
For figure 3 and figure 6, we also checked again carefully. But here all respective loading controls were presented already before. Therefore, we modified the orientation of the shown protein bands in the respective figures and also included, hopefully more reliable raw data in the additional document. This new raw data demonstrates that the positioning of the membrane during detection was different and therefore, also the visualization within the figure.
Additionally, as we changed our protein detection system in between of the project, from (I) detecting Hyperfilm only with chemiluminescence to (II) detection with a Computer-based system without films, we now included in the additional document by which system each membrane was detected. This will also explain different presentations of raw data. We also included the fact of using different detection systems in the material and method section.
Reviewer 2 Report
The authors have made the corrections appropriately.
Author Response
We thanks Reviewer 2 for the time to review our manuscript again and are happy that we have convinced the reviewer already with our first revision. We hope that also our second revision is in line with reviewer 2 comments before.